# Breeding Population and Nesting Habitat of Skuas in the Harmony Point Antarctic Specially Protected Area

Francisco Santa Cruz [1,*] and Lucas Krüger [1,2]

[1] Departamento Científico, Instituto Antártico Chileno, Plaza Muñoz Gamero 1055, Punta Arenas 6200000, Chile
[2] Millennium Institute Biodiversity of Antarctic and Subantarctic Ecosystems (BASE), Las Palmeras 3425, Ñuñoa 7750000, Chile
* Correspondence: fsantacruz@inach.cl

**Abstract:** Regular monitoring of breeding population abundance and environmental factors related to the nesting habitat has proven fundamental for seabird conservation. Harmony Point (Nelson Island) is an Antarctic Specially Protected Area (ASPA) designated to manage and protect the high biological value of seabirds' richness and abundance. However, due to the remote location of their breeding sites, many species lack updated population counts. Skuas (*Catharacta* sp.) exhibited a two-decade gap since the last census was conducted in Harmony Point. The abundance and spatial distribution of the nests of skuas were studied during the austral summer of 2019/2020. Through an exhaustive search, we counted and mapped active nests. Nesting habitat was assessed by the use of an unmanned aerial vehicle to take aerial pictures and build an orthomosaic image to determine vegetation in the area. Additionally, a digital elevation model was built to calculate a series of geomorphological-related variables. Suitability models were used to estimate the importance of variables to the nesting of skuas. A total of 71 brown skua (*Catharacta antarctica*) and 3 south polar skua (*C. maccormicki*) nests were counted. Two of the seven variables (terrain slope and vegetation cover) accounted for 57.5% ± 14.1% of the models' variability; sun radiation incidence, and wind shielding were of secondary importance. Water flow accumulation, distance from penguin colonies, and terrain elevation were the least important variables. Skuas selected for nesting flat terrains (slope < 10°) with a vegetation cover of above 20%, slightly higher sun incidence (270 to 280 kW/h), and intermediary windshielding (45% to 55% of exposition). Considering previous estimates, the skua species at Harmony Point has kept an apparently stable population size over the last 25 years. However, expected changes in nesting habitat availability, i.e., increased snow-free area, increased wind intensity, changes in vegetation cover, and reduction of penguin populations, might change population size in the mid to long term.

**Keywords:** *Catharacta*; brown skua; south polar skua; habitat suitability; South Shetland islands; topography





## 1. Introduction

There is wide recognition of the great importance of seabird conservation in the Southern Ocean. Great efforts have been conducted to establish and maintain protection in zones [1,2], especially in important breeding areas [3] where human activities and environmental changes related to climate change can have large negative impacts [4,5]. The isolation and limitations of accessing Antarctica makes it difficult to maintain regular monitoring to visualize population trends and detect the environmental factors involved [6]. Usually, the climatic conditions and logistic constraints make the effort to survey the species limited and restricted to certain species of greater interest or charismatic ones (i.e., penguins), leaving others without updates for decades [2]. This is particularly problematic for those species considered to be top predators, whose population abundances are comparatively lower to medium and have low trophic levels, and population dynamics can be particularly sensitive to rapid environmental change [7–9].

Skuas are long-lived top predators widely distributed in sub-Antarctic and Antarctic environments [7,8], whose population dynamics can be representative of ecosystem health [10]. The brown skua (*Catharacta antarctica*) and the south polar skua (*Catharacta maccormicki*) are two species commonly found breeding sympatrically in the Antarctic during summer [11]. During the spring, adults migrate from winter locations in temperate areas to sub-Antarctic and Antarctic sites to breed. Adults prepare the nest, lay up to two eggs and perform parental care during the chick-rearing period [12]. Skuas play a key role as opportunistic predators with highly territorial behavior around their nesting areas [13]. Skuas mostly feed their young with penguin eggs and chicks, and breeding success depends on a sufficient amount of nearby food supply [14]. Nest site selection is therefore commonly influenced by proximity with penguin colonies [14]. However, there are also geomorphological features such as terrain elevation, composition or exposure to sun, wind and rain, that set nest habitat in order to provide conditions for the development of the chicks [11,15,16].

At Harmony Point (Nelson Island, South Shetland Islands), significant work has been done to maintain population estimates of seabird species, although with large temporal gaps [17–19]. The last count of skua breeding pairs was conducted in 1995 [17]. There is also scarce information about nest distribution, nest availability, and suitability associated with environmental factors. The objective of this work was to update the number of breeding pairs of skuas at HP, study the spatial nests distribution, and through an unmanned aerial vehicle (UAV) survey and identify geomorphological characteristics where the nests are located.

## 2. Materials and Methods

### 2.1. Nest Mapping

The skua nests' distribution was surveyed between 26 November 2019 and 15 January 2020 in Harmony Point (Figure 1a,b). An exhaustive visual search by walking throughout the ice-free area was conducted (an accumulated linear distance of 129.6 km was explored, Supplementary Material Figure S1), and active nests (a nest with an adult in defensive behavior), when detected, were marked with a handheld GPS receiver from a distance of approximately 50 m from the nest (to reduce interference over the breeding bird). The identification of skua species was performed according to morphological descriptions of plumage coloration and body appearance (the south polar skua is smaller with golden hackles, and the brown skua is larger and heavy with white spotted hackles at the back) [20,21].

### 2.2. Nest Habitat

During January 2020 we performed a flight survey using the UAV DJI Mavic 2 Pro (DJI, Shenzhen, China), equipped with a 20MP RGB Hasselblad L1D-20c camera. The UAV was flown at a horizontal speed of 5 m/s following parallel transects at an altitude of 400 m in order to achieve consistent sets of imagery and proper overlap between images. Single photographs were taken every 5 s, covering all of the ice-free area. Photographs were processed in Agisoft PhotoScan Professional (version 1.2.6) in order to create a high-resolution georeferenced orthomosaic image (pixel size ≈ 20 cm) and a digital elevation model (1 m cell size). The orthomosaic is a highly detailed and undistorted map built with the UAV imagery that enhances the visibility of surface details in the surveyed area, being especially useful in hard-to-access study systems [22].

Vegetation cover was estimated by applying a maximum likelihood unsupervised image classification of the orthomosaic on ArcMap 10.8.2. Unsupervised algorithms for image classification find the underlying structure of the image automatically by clustering data which is spectrally similar [23]. Such methods are useful for organizing large and spatially complex sets of data efficiently and accurately [24]. Vegetation-covered cells were set to value of 1 and other types of substrates were set as zero. The grid was then resampled to 1 m × 1 m using a mean value, therefore, the final value was a proportion of a 1 m$^2$ cell that was covered by any type of vegetation (Figure 1c). At the elevation the

drone was flown, mosses and algae were not separated, therefore they are being considered together. By observation on the field, we can say that moss carpets are dominant as large vegetation formations, and algae can be found mostly downstream of large penguin colonies, representing a small proportion of "green cells" in the orthomosaic. Large lichen fields at the southeast of the area were separated by the unsupervised algorithm (Figure 1c).

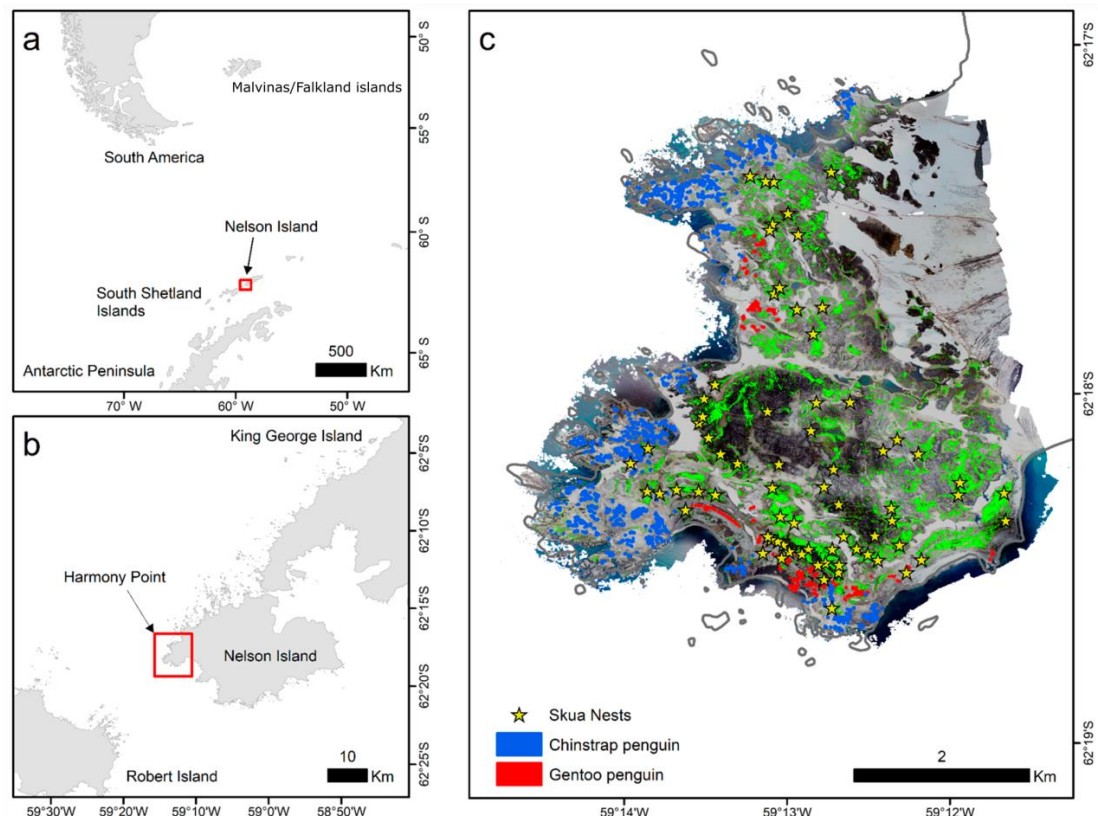

**Figure 1.** Locations of Nelson Island (**a**) and Harmony Point (**b**); Harmony Point in detail showing the orthomosaic built from aerial imagery taken on 400 m altitude, the classified vegetation cover (bright green) the mapped skua (*Catharacta* sp.) nests (yellow stars), and the penguin colony polygons (**c**). Map created using ArcMap 10.8.2 (https://www.esri.com/, accessed on 10 January 2023). Land contours from Esri Global Mapping International (https://www.arcgis.com/; item id a3cb207855b348a297ab85261743351d, accessed on 10 January 2023) and the SCAR Antarctic Digital Database 2022 (https://www.add.scar.org/, accessed on 10 January 2023).

A digital elevation model DEM (Figure 2a) was the basis to generate four other topographical variables: terrain slope, incidence of sun radiation, windshielding, and water flow accumulation that were processed through ArcMap 10.4 toolboxes (ESRI 2022). Terrain slope (Figure 2b) was determined by applying the surface tool, and is a measure of the inclination of a cell estimated through the elevation difference from neighbor cells on a grid. The Area Solar Radiation tool was used to calculate the sun radiation incidence SUN (Figure 2c), using multiple days from 1 December 2019 to 31 January 2020. SUN is measured as kilowatts per hour (kW/h) reaching the surface in that period. Windshield (Figure 2d) was estimated using the hillshade tool, creating a shaded relief from a surface raster by considering the illumination source angle and shadows. Multiple outputs were generated using predominant wind directions for the area as the source (azimuth) angles (270°, 300°, 330°, 0°, 30°, as west and north winds have been more frequent on the area over the years, authors pers. inf.) and with multiple incident altitude angles (5°, 10°, 15° and 45°). A mean value was calculated for all of those outputs. Hillshade indicates the amount of shaded (0) and lightened (255) relief, therefore, mean hillshade values were divided by 255, therefore indicating, in terms of proportion, the level of "protection" the

relief provides from the "source", in this case, the predominant wind direction. The flow accumulation (Figure 2e) tool uses a measure of flow direction based on the terrain and the elevation slope to accumulate the weight of all cells that flow into each downslope cell. The final output was classified in 0 (low flow) and 1 (higher flow), and a mean filtering (5 × 5 cells) was used to average the values, therefore, generating a measure of water accumulation probability.

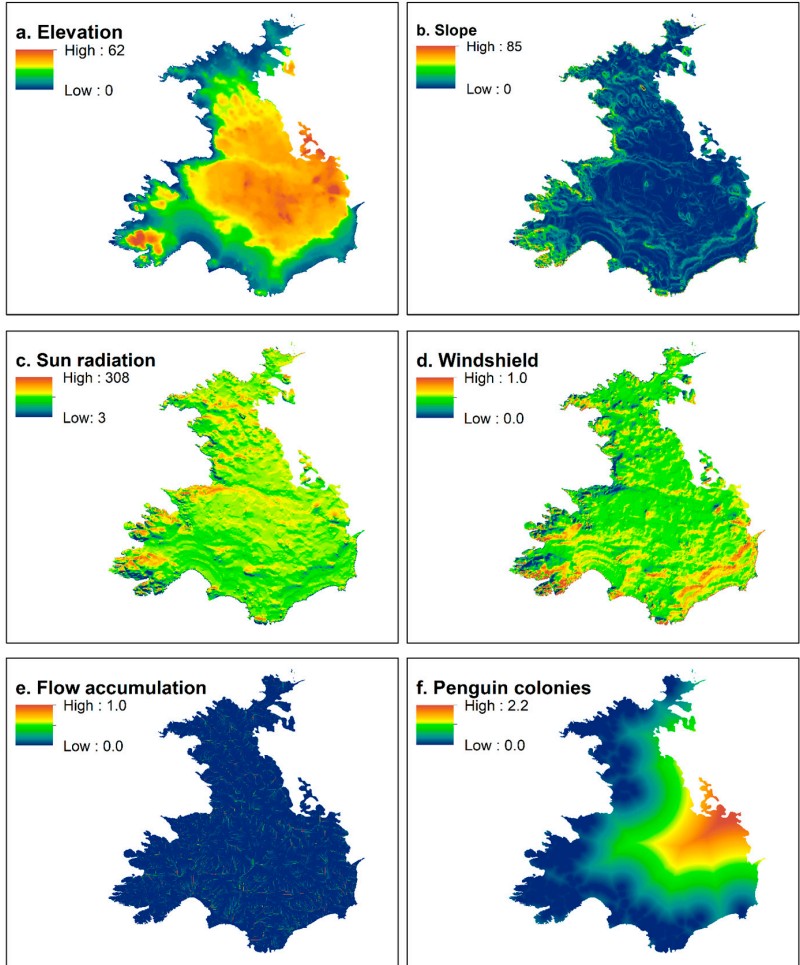

**Figure 2.** Habitat variables used for applying nesting habitat suitability models using data of skuas (*Catharacta* sp.) breeding in Harmony Point, Nelson Island: terrain elevation (**a**), terrain slope (**b**), sun radiation incidence (**c**), windshield (**d**), water flow accumulation (**e**), and distance from penguin colonies (**f**).

The orthomosaic image (Figure 1c) also allowed for the identification of penguin colonies, which we used to construct polygons marking the edge of the colonies. The Euclidean distance tool was used to calculate the horizontal distance between nest and penguin colonies (Figure 2f).

In order to evaluate habitat preferences, 300 random points were generated throughout Harmony Point to use as pseudo absences (see next section). To sample environmental variables, a 100 m buffer was generated around each geographical position (nests and random points). That procedure was used as: (i) the position of the skua nests were marked from a nearby position, (ii) chicks are mobile soon after they hatch [25,26], and (iii) breeding skuas defend territory around their nests [13,27]. Therefore, nesting habitat is better described by the amount of variables around the nest, rather than considering values on a single position. Habitat variables were averaged within each of those 100 m-buffers to be used in subsequent analysis.

*2.3. Nesting Habitat Suitability*

Nests' positions and random points were used to run ecological niche models (ENM), also called habitat suitability models (regarded here as nesting habitat models). Those models use species' presence to estimate habitat suitability based on a set of environmental variables, calculating a probability (0 to 1) that the available habitat on the landscape is similar to the observed occurrences [28]. While several ENM methods exist, some of the more accurate and robust methods use machine-learning iterative approaches to train and test models over a subset of the original data [29–31]. Those methods assume species absences are unknown, but require the use of absences to estimate probabilities, therefore the generations of pseudo-absences (points where the current dataset have no presences but true absence is somewhat uncertain) are required [32,33]. Random points, described in the previous section, were used as pseudo-absences.

Generalized boosted models (or Gradient Boosting Machines) GBM were used to apply ENM and calculate nesting suitability based on the probability of nest occurrence in relation to the environmental variables. GBM [34–36] is a machine-learning based technique which, used in the context of ENM, fits occurrence of a species (binary data; Bernoulli distribution) to continuous variables through applying iterative regression trees and using a cross-validation selection procedure to keep the best models and optimize prediction [36]. GBM is regarded as one of the most robust predictive statistical techniques [31,37]. The 'gbm' R-package was used [35]. GBM was applied 30 times using the nests as presence and a random subset of pseudo-absences matching the number of nests in order to have a balanced estimation of probabilities that are also independent of the pseudo-absences' generation process [33].

The following parameters were used: 50 regression trees (number of iterations); a bag fraction of 0.8 (proportion of the training set used to propose the next tree); a shrinkage of 0.1, as the learning rate or the contribution of each tree to the next one; the train fraction was 0.8, that is, 80% of data were used to train the model, and the remaining 20% to test accuracy; a tree depth (number of interactions) of 1 (no interaction); and the minimum number of observations in terminal nodes = 10. Performance and accuracy were measured through three methods: cross-validation error, that is, the average error of the testing fraction (20% of original data set selected on a 10-fold cross-validation); the out-of-bag error OOB (the reduction in prediction deviance comparing sequentially iterated trees); the area under the receiver operating the characteristic ROC curve (AUC). AUC is calculated from the sensitivity (true positive rate) and specificity (true negative rate), which measures proportions of correct classifications through the model outputs. Values of AUC near 0.8 are considered satisfactory and above 0.9 are good (for further details on AUC see [38]). Variable contributions to the nesting habitat were calculated by measuring the percentage of change in model performance when the variables are taken off the model during the 10-fold cross-validation procedure.

The GBMs runs were used to predict suitability in the 377 geographical positions (random points plus nests), and a mean value for each grid cell was calculated out of the outputs of the 30 runs. The value was posteriorly interpolated to a regular grid using an Inverse Distance Weighting approach. Suitability was grouped into five regular classes since a threshold value of 0.50 (below this value most variables have no effect on habitat selection): unsuitable (<0.50), low suitability ($\geq$0.5, <0.60), intermediary suitability ($\geq$0.60, <0.65), high suitability ($\geq$0.65, <0.70), and optimum stability ($\geq$0.70).

For detailed methods and codes, please see File S1.

## 3. Results

By walking linearly for a total of 129.6 km we covered the entire 18.4 km$^2$ of ice-free area in Harmony Point. We found 74 active nests (Figure 1c). Of those, only three nests were from south polar skuas. The nests presented a clustered distribution ($z = -4.07$, NNR = 0.753, $p < 0.001$) with an observed mean distance of 182.45 m between nests.

### 3.1. Nest Habitat Model Performance and Variable Contribution

The model's accuracy was satisfactory (0.79 ± 0.01). The increasing iterations were able to approximate deviance to 0 (Figure 3a) and reduce errors (Figure 3b,c) for all 30 models run. Geomorphological variables (Figure 3d) with a higher importance on the models were terrain slope (31.92% ± 9.45%, 18.74% min, 55.81% max) and vegetation cover (25.55% ± 9.06%, 15.24% min, 56.26% max). Windshield (10.37% ± 7.95%), sun radiation incidence (10.20% ± 7.86%), and flow accumulation (9.26% ± 8.67%) had a secondary importance. Distance from penguin colonies (8.03% ± 7.00%) and terrain elevation (6.07% ± 6.00%) were the least important variables.

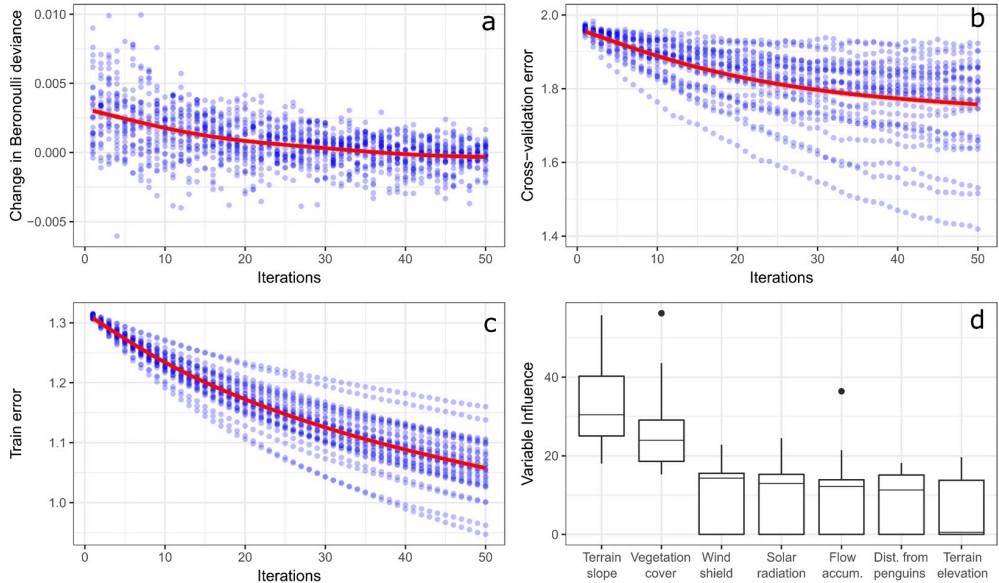

**Figure 3.** Generalized boosted model's performance statistics at each of the 50 iterations for the 30 model runs showing an approximation of prediction deviance to 0 (**a**) and reductions of cross-validation (**b**) and training (**c**) errors at each new iteration. Frequency distribution of environmental variables influence on models output measured as percentage of decrease in classification accuracy by elimination of the variable from the model (**d**). Estimated values from the GBM runs and iterations (blue dots) fitted automatically with a general additive model (GAM) trend (red line). Black dots are boxplot outliers.

### 3.2. Nesting Habitat

Skuas preferred to nest on flat terrains (when slopes were above 10° of inclination suitability moved below 0.5; Figure 4a) with intermediary to high vegetation cover (there was a lot of variability on suitability when vegetation cover was below 20%, but above 20% suitability tended to be always higher than 0.5, Figure 4b). Intermediate values of windshield were preferred for nesting (Figure 4c) and higher values of sun radiation incidence predominated with suitability >0.5 (Figure 4d). There was a tendency for nests to be placed on slightly higher flow accumulation areas, but also with a large variability (Figure 4e). While skuas nested in areas far from penguin colonies, there was a recorded concentration of higher suitability values on distances below 500 m from penguins (Figure 4f). Elevation did not have any important contribution nor do response curves have any detectable trend. The majority of nests were placed in areas of optimum suitability (56.6%), especially concentrated at the southerly sector of Harmony Point (Figure 5), in opposition with zones of high suitability (12.2%) and intermediary or low suitability (29.7%). Only one nest was positioned in an area classified by the model as unsuitable (Figure 5).

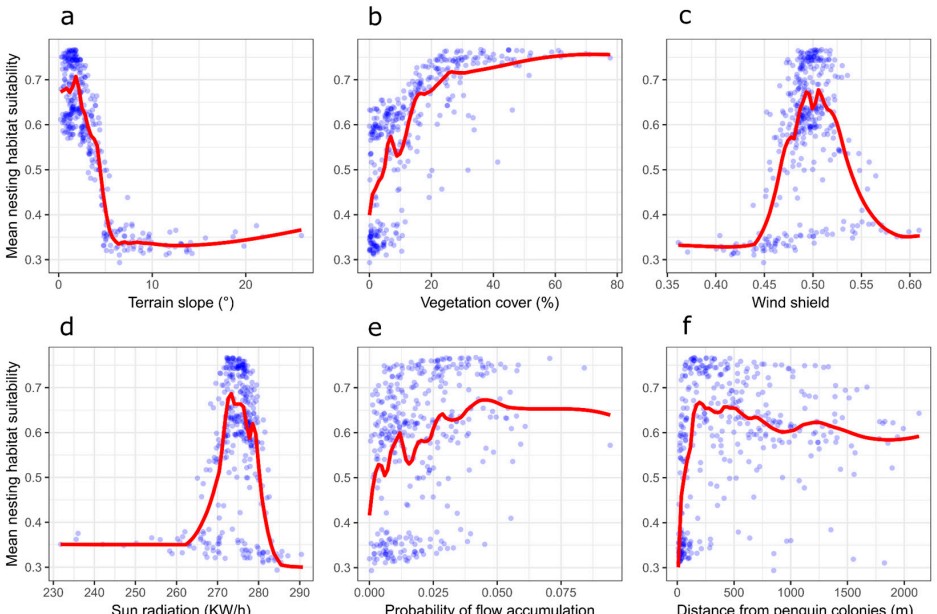

**Figure 4.** Mean Skua (*Catharacta* sp.) nesting habitat suitability (blue dots) predicted from the 30 Generalized Boosted Model runs over the Harmony Point (Nelson Island) area in relation to the environmental variables of terrain slope (**a**), vegetation cover (**b**), wind shield (**c**), sun radiation (**d**), flow accumulation (**e**) and distance from penguin colonies (**f**). Red lines are automatically generated general additive model (GAM) trends.

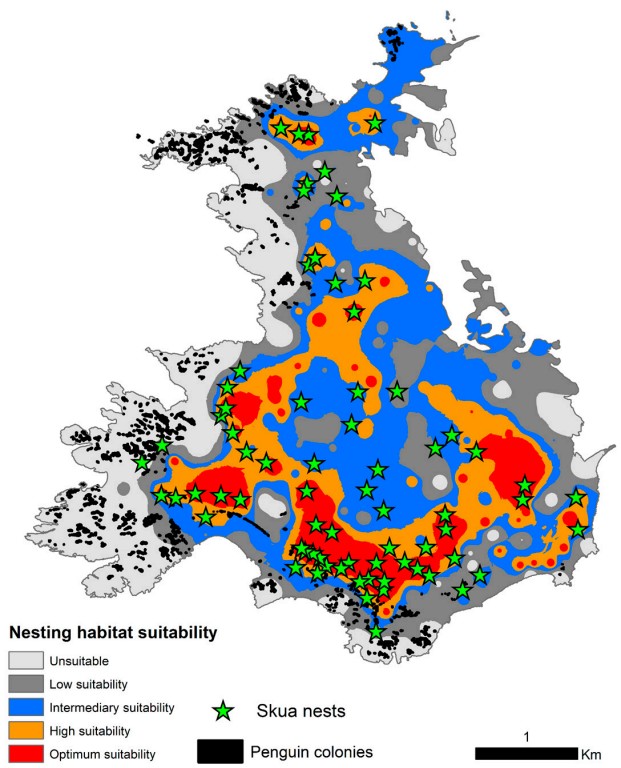

**Figure 5.** Estimated nesting habitat suitability for skuas (*Catharacta* sp.) in Harmony Point, Nelson Island. Suitability classes were assigned as Unsuitable (<0.50), Low Suitability (≥0.5, <0.60), Intermediary Suitability (≥0.60, <0.65), High Suitability (≥0.65, <0.70) and Optimum Suitability (≥0.70). Mapped nest skuas (green stars) are plotted with the distribution of penguin colonies (black polygons).

## 4. Discussion

Skuas' nest distribution across the ice-free areas of Harmony Point was successfully modeled using geomorphology features and penguin–colony distance. As skuas are a species that defend a territory both for breeding and feeding [39], these factors may be important factors in nest-site selection. The generalized boosted model exhibited a satisfactory performance with low deviance and error, with terrain slope and vegetation cover being the most important variables. These findings are consistent with the breeding description that skuas commonly nest in highlands of low slope with a large presence of vegetation carpets [40].

We observed that nesting suitability increased with the increase of vegetation cover, especially at snow-free sites covered with moss-turf carpets [21]. Carpets provide a safe space where eggs and chicks are perfectly camouflaged (Figure 6). This could also be supported by the relation with accumulation flow, as mosses require a minimal level of water availability to subsist [41]. One study on another skua species has shown that adults make frequent use of ponds, water streams, and shallow lakes to clean feathers after feeding events [42], which, by field observation, seems to be the case for brown and south polar skuas.

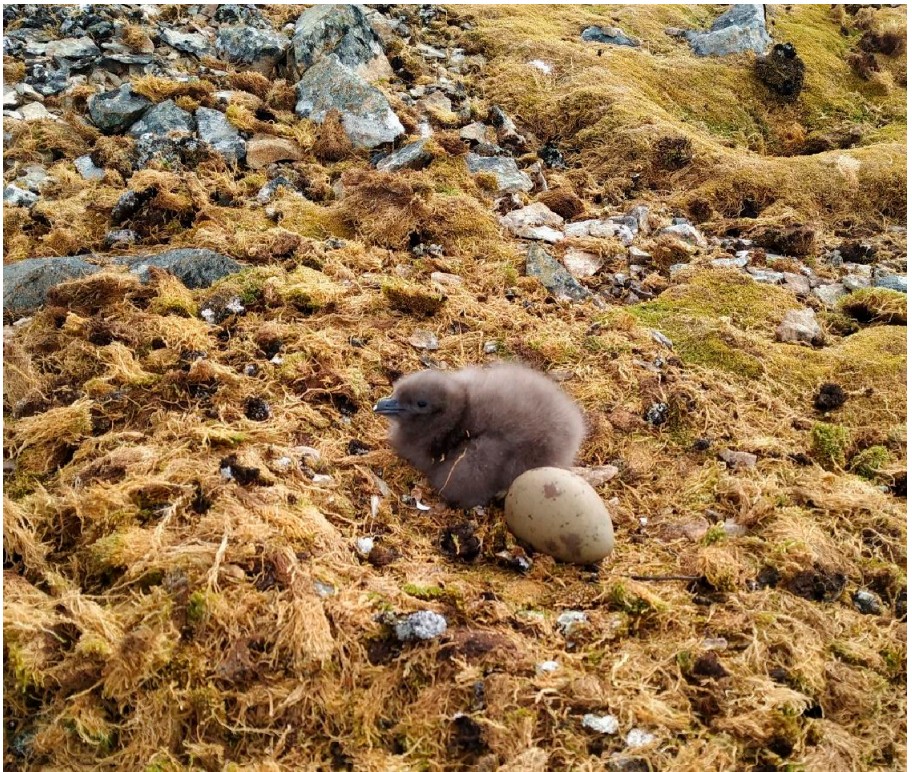

**Figure 6.** Skua nest on moss-turf carpet at Harmony Point. The color pattern of the egg and chick allows a perfect camouflage. Photograph taken by Francisco Santa Cruz.

Nesting suitability was higher in sites with slopes <10°, which is similar to information observed for skuas breeding at Cierva Cove [40]. Slope is a factor closely related to windshield, sun radiation incidence, and flow accumulation being associated with local fine-scale variation in thermal and wind stress, relevant for nests' protection [19]. Skuas' nesting occurred at intermediate values for those geomorphology features, suggesting that skuas avoid extreme conditions. Intermediate values could be representatives of a high-quality nesting habitat, providing optimum conditions of temperature, humidity, and shelter for eggs and chicks. This is particularly important in the Antarctic environment, where nest characteristics are key to face the frequent strong winds and snow during the

breeding season which can affect breeding success [43], particularly for skuas whose chicks are nidifugous within a few days after hatching [25,26].

We updated the breeding population of two skua species at Harmony Point, where according to the previous counts [17], we observed a difference of active nests from 61 to 71 for the brown skua, and from 10 to 3 nests for the south polar skua. Considering the two species together, the values of the 71 nests registered in 1995/1996, and 74 in 2019/2-20 suggest stability. However, more nest counts are necessary in order to support that claim and rule out the effects of high inter-annual variability on breeding numbers. Other colonies in the sector, such as Potter Peninsula, have shown stability [44], however, decrease has also been seen in areas where the number of potential prey have decreased [45,46].

There are contrasting hypotheses in previous studies on the interaction, competitiveness, and displacement between the two species of skua. Some suggest that the increase in population of the south polar skuas may displace the brown skuas [13], while others indicate that due to their greater predatory ability, the brown skuas are able to displace the south polar skuas [47]. At Harmony Point, the decline in population of the south polar species could lead us to believe that the latter is the case. However, again, we do not have enough data to conclude that this is only the result of inter-annual variability on breeding numbers.

The apparent breeding population stability of skuas at Harmony Point suggests that the colony is in a steady state where the carrying capacity or saturation level has been reached, as has been pointed out for other skua colonies in Antarctica [44]. For seabirds, the carrying capacity of a species is related to resource limitation including food availability and habitat availability [48]. For instance, in HP we assume food is not a limitation, as the chinstrap penguin (*Pygoscelis antarcticus*) colony can surely provide enough food for skuas, especially considering that its diet is centered on penguins. Unlike in other areas, there is a greater contribution of other seabird species, fish, and other marine resources [49]. Consequently, we argued that skuas in Harmony Point have optimized the use of available nesting habitats, particularly in terms of space and maximum nest density. According to our results, the areas of optimum habitat suitability are densely occupied (Figure 5). We calculated a nest density of 5.8 nest per $km^2$ (74 nests on the 12.8 $km^2$ of estimated suitable area) which is comparatively low compared to other areas with up to 132 pairs per $km^2$ [50], however, we also estimate a mean of 182.45 m between nests, similar to that reported in Admiralty Bay, where skuas that breed close to penguin colonies have a strategy of keeping distance to avoid cannibalism [11]. It is important to note that skuas defend their eggs and chicks from predators, including cannibalism from neighbor breeding conspecific [39], therefore, the maximum nest density might be influenced by a minimum distance to avoid overlapped territories.

The interaction of geomorphological features combined with the density of conspecifics and food resource availability may modulate skuas' nest-site selection, therefore, breeding success and population stability. Harmony Point is a sensitive area representative of the invaluable richness of Antarctic seabirds, protected since 1985 as an Antarctic Specially Protected Area, thus suffering little disturbance. Here, any type of driver controlling the nest and breeding ecology has responded to natural processes with minimum direct human-related pressures. Although the chinstrap colony is still numerous and capable of providing food, its abundance is likely shrinking (see [51]) and the carrying capacity in coming years may not be influenced only by space availability, but also by food availability and its combination. Thus far, ASPAs have proven to be an effective regulatory mechanism but it also may depend on the spatial scale considered. At a local scale, it avoids direct human intervention on seabird populations and surely has allowed the population stability of skuas in Harmony Point. However, the increased pressure from human-stressors related to the presence of scientific activities [52], debris pollution [53,54], diseases dissemination [55], alien species arriving [56], and climate change [57], are now challenging its effectiveness. ASPAs might not be effective when large-scale processes occur. For example, the decline of chinstrap penguin abundance in the South Shetland islands has been associated with

climate-change induced regional reductions in winter sea–ice cover that impact primary production and consequently reduce the productivity and availability of Antarctic krill [57]. In that context, only 17% of ASPA's management plans have been updated to include issues related to climate change [58]. It is a priority to expand these updates but also relevant to rethink how ASPAs can still be effective protection tools, when climate-related changes impact on scales beyond the scale of the ASPA itself. Large-scale tools focused to protect ecosystem processes, such as Marine Protected Areas [59], arise as comprehensive initiatives aimed at providing spaces without extra pressures to increase the resilience of natural populations.

## 5. Conclusions

Our results show that, according to previous estimates, the two skua species at Harmony Point have kept an apparently stable population size over the last 25 years. Skuas show that the optimal suitability for nesting habitat has been used almost entirely, composed mainly of sites with high vegetation cover, low slope, and intermediate values of exposure to wind and solar radiation.

**Supplementary Materials:** The following supporting information can be downloaded at: https://www.mdpi.com/article/10.3390/d15050638/s1, Figure S1: Sampling effort for searching for skuas nests in Harmony Point between 26 November 2019 and 15 January 2020. A hand-held GPS receiver was set to conduct an automatic background tracking while researchers searched for nests, recording in its internal memory one geographical fix each 5 seconds. File S1: SantaCruz_and_Kruger_DataAndScript.

**Author Contributions:** Conceptualization, F.S.C. and L.K.; Formal analysis, L.K.; Methodology, F.S.C. and L.K.; Writing—original draft, F.S.C. and L.K.; Writing—review and editing, F.S.C. and L.K. All authors have read and agreed to the published version of the manuscript.

**Funding:** This study was supported by the Instituto Antártico Chileno through the Programa de Áreas Marinas Protegidas (AMP 24 03 052). L.K. is also partially funded by the ANID-Millennium Science Initiative Program—ICN2021_002.

**Institutional Review Board Statement:** The entry to the Antarctic Specially Protected Area of Harmony Point (ASPA N°133) was authorized by the Chilean Antarctic Institute through permits 1045/2019 and 1105/2019, in accordance with Decision 1 (2002) of the XXV-ATCM on the system of Protected Areas, and the provisions of Annex V of the Madrid Protocol.

**Data Availability Statement:** The data presented in this study are available as RData file and R code of analysis.

**Acknowledgments:** The authors thank the crew of the Chilean army OPV Marinero Fuentealba, and the crew of R/S Karpuj of the Chilean Antarctic Institute for logistical support to access Nelson Island.

**Conflicts of Interest:** The authors declare no conflict of interest.

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
