# Peer review of "Breeding Population and Nesting Habitat of Skuas in the Harmony Point Antarctic Specially Protected Area"

_diversity, doi:10.3390/d15050638_

Round 1

Reviewer 1 Report

Review for Diversity Manuscript ID: diversity-2354839

Article: Breeding population and nesting habitat of skuas in the Harmony Point Antarctic specially protected area

Comments to the Author

This paper deals with a subject that would be of interest to readers of Diversity and also deals with a subject that excites controversy within the seabird conservation.

In short, this is a nice paper. I think that is an interesting and useful contribution to the literature about seabird conservation and specifically species distribution modelling. In general, although I don't have any experience in using Generalized boosted models, the writing is clear. The arguments are clearly presented, the results are interesting and the interpretation of the results justified.

I have never worked with seabirds. My extensive experience has been with direct work with land birds where handling caused them no problems. I don't understand why these birds are so sensitive. They are not supposed to have interference with humans, so it would be logical to think that they are not afraid and will not abandon their nests. My comment is along the lines that it would be interesting to assign the reproductive success of each pair to the habitat category that has been assigned to see if there is any correspondence according to it. Another thing that strikes me is the low presence of the other Skua species. It would be interesting to see the inter-annual variation, through longitudinal studies over time, with tagged individuals to see the recruitment rate in this species. 

Minor comments:

-       Line 50. I find it difficult to think that it is necessary to maintain a 50 m margin to take the position of the nest with the GPS.

-       Line 136-162. As I have commented before, I do not have the experience to evaluate whether this paragraph adequately describes the technique used. It seems fine to me....

-       Line 182. I think the authors were very anxious to finish the article.....:). The end of the sentence is missing. I think it should read: “…were the least important variables”.

-       Line 200. Replace “harmony point” for “Harmony Point”

-       Line 349. Remove the title in capital letters.

-       Line 378. Remove the title in capital letters.

Sorry for my low level of written English.

Reviewer 2 Report

This is a useful and relevant paper that provides some quality information on skua distributions as relevant to conservation initiatives. Please check the written English throughout and please consider the description of your data analysis to ensure it is repeatable for others to follow. 

The abstract needs to start with a sentence or two that provides background information to the research question. 

For numbers less than ten, please write out in words (unless they contain units, e.g. 2cm). Please check throughout the manuscript.

Please can you provide a bridge between line 38 (the general information) and then the text on skuas. Why these species? What focus?

When a species with the genus name as one previously used in the same part of the text is included, it can be abbreviated to just the first letter of the genus name. E.g. line 39 and 40. Please check throughout manuscript.

Line 43: Suggest "Skuas mostly feed their young by preying on penguin eggs and chicks."

Line 44: define proper food supply

The ecology of skuas in the introduction is very brief. Do you need to include any further information on the natural history of these species to set the scene for the paper?

Line 65: their morphological... And perhaps give a brief table of what these are?

Figure 1: Copyright of the maps? Do you need to state in the text?

(and for all other maps- do you need to provide a reference or copyright?)

Line 114: please define what an orthomosaic is

Line 118: Please explain this as it is hard for the reader to follow.

Line 119: Please do not start a sentence with Because 

Line 123: What is posterior analysis?

Please can you start your data analysis section with the stats package that was used

Line 152: what is out-of-Bag error? Please define.

Line 161: What is mean estimated suitability?

 You have clearly undertaken some very complex analyses but please can you make sure the reader knows exactly what was texted and why? 

Figure 3: Please can you explain how these figures show the improvement in model fit?

I think Figure 6 needs to be moved out of the results section. 

Please check throughout use of skua and skua for plural. Sometimes you use skua and sometimes skuas. Consistency would be good here. 

Line 267: chinstrap penguin (and give scientific name)

Line 272: important to note

Line 285-287 does not feel like the end of the discussion. Please can you provide some interpretation to these discursive points before you move on to your conclusions. 

If the chinstrap penguin colonies do shrink in productivity, what is the best conservation action for the skuas? Can you comment or suggest practical initiatives? 

Please see my report

Reviewer 3 Report

I have completed my review for ‘Breeding population and nesting habitat of skuas in the Harmony Point Antarctic specially protected area’, which is currently under consideration for publication in Diversity. Here, the authors estimated and characterized the spatial distribution of nests of skuas at Harmony Point. Also, they developed habitat suitability models for the nesting of skuas. The skua populations appear stable, and the most important habitat variables for nesting were terrain slope and vegetation cover. I enjoyed reading the manuscript and believe it could be an essential addition to the updated population estimates and habitat suitability of the Antarctic seabird species. In particular, when environmental change may influence habitat availability and suitability for nesting skuas. However, the manuscript has some issues that need further clarification.

Adding information about each skua species' breeding biology, such as clutch size, would be beneficial. Also, given the highly territorial behavior that characterizes the species, it is important to tease apart what factors are affecting the territory size and then the location of the nest. The mean distance between nests is relatively short (less than 200 m), so the nest density at the study site is relatively high compared with other sites. Moreover, the apparent stable population of skuas at Harmony Point may be the consequence of the carrying capacity of the study site for skuas territories rather than the nest suitable habitat. It would be beneficial if the authors discussed these scales of the spatial organization of breeding skuas, territory, and nest.

The accuracy of the nest habitat model was satisfactory, and the terrain slope and vegetation cover were the most important for the model. However, the image classification of the vegetation cover was unsupervised. Is it possible that this unsupervised classification could produce a bias in the habitat model performance? This could be potentially important when the difference between suitability classes is slight. For example, the difference between Intermedium and Optimum suitability is only 0.10. In any case, the authors can explain the suitability classes criteria further.

On page 4, line 119. It should be “section” instead of “session”.

On page 5, lines 181-182, it seems like an unfinished sentence.
